# Detecting the Presence of Intrusive Drilling in Secure Transport Containers Using Non-Contact Millimeter-Wave Radar

**DOI:** 10.3390/s22072718

**Published:** 2022-04-01

**Authors:** Samuel Wagner, Ahmad Alkasimi, Anh-Vu Pham

**Affiliations:** Department of Electrical and Computer Engineering, University of California, Davis, Davis, CA 95616, USA; aalkasimi@ucdavis.edu (A.A.); ahpham@ucdavis.edu (A.-V.P.)

**Keywords:** millimeter-wave radar, non-destructive sensing, security

## Abstract

We employ a 77–81 GHz frequency-modulated continuous-wave (FMCW) millimeter-wave radar to sense anomalous vibrations during vehicle transport at highway speeds for the first time. Secure metallic containers can be breached during transport by means of drilling into their sidewalls but detecting a drilling signature is difficult because the large vibrations of transport drown out the small vibrations of drilling. For the first time, we demonstrate that it is possible to use a non-contact millimeter-wave radar sensor to detect this micron-scale intrusive drilling while highway-speed vehicle movement shakes the container. With the millimeter-wave radar monitoring the microdoppler signature of the container’s vibrating walls, we create a novel signal-processing pipeline consisting of range–angle tracking, time–frequency analysis, horizontal stripe image convolution, and principal component analysis to create a robust and powerful detection statistic to alarm if drilling is present. To support this pipeline, we develop a statistical model combining the vibrating container and the random vibrations induced by vehicle movement to explore the robustness of the sensor’s detection capabilities. The presented results strongly support the inclusion of a millimeter-wave radar vibration sensor into a transport security system.

## 1. Introduction

Both government and industry have a growing need to develop contactless autonomous observation systems for the security of transport containers. Transport containers often carry sensitive material. A common method of attack is a drill attack—intruders drill into the containers either to insert an endoscope to view the inside or to obtain the secured material. To further complicate detection, the drill attack may happen while the container is in transport. Several methods in the literature address the problem of container intrusion. For example, Ref. [1] uses vibro-converts to force vibrations in order to create a unique acoustic fingerprint for each container, which shows significant changes if any physical alteration occurs. Moreover, Ref. [2] employs a three-axis acceleration sensor mounted on the container to observe intrusions using a microcontroller and a communication module. An approach based on microphones and neural networks was shown in [3] to detect the sound profile generated while attempting to cut open the containers. However, all proposed methods may either not be sensitive enough for weak drilling during transport or require contact with the container, which is undesirable from a security point of view. To solve the problems in current literature solutions, we propose using a millimeter-wave radar to monitor the vibration of a transport container.

In this work, we applied a commercially available millimeter-wave frequency-modulated continuous-wave (FMCW) radar system to the problem of detecting intrusive drilling during transport. The millimeter-wave automotive radar is a good choice for this sensing application because the automotive radar revolution has resulted in low-cost and widely available commercial options. The millimeter-wave radar has a small enough wavelength to measure down to micron-level vibration at a stand-off distance, can monitor wide spatial areas using multiple receive and transmit antennas, and can achieve high reliability. The FMCW radar technique is used in place of other techniques such as continuous-wave (CW), stepped-frequency continuous-wave (SFCW), or impulse because FMCW allows for excellent range discrimination due to bandwidth, chirp acquisition time, and an acceptable signal-to-noise ratio (SNR). FMCW radars have been previously used to address problems related to vibration and small range variation estimation, including the non-contact range tracking of vital signs [4], wide area surveillance for perimeter protection [5], and observing the vibration in infrastructure such as bridges or buildings [6]. Another option to measure vibration is a laser Doppler vibrometer (LDV), which is also used in this work to verify spectra. The LDV requires line-of-sight to the container and cannot automatically track a wide area if the container is moved or is misplaced. In addition to being inexpensive, a millimeter-wave radar’s antennas can penetrate light cover and monitor a sizeable swath of space, making the millimeter-wave FMCW radar’s operation more robust than an LDV’s. 

In this work, we present a radar-based surveillance system with the primary advantage of being autonomous, contactless, and having a wide field of view and long range. The system is able to detect intrusive drilling in transit scenarios at highway speeds of 112 kph. First, we provide a brief theoretical background on the operation of the FMCW radar with respect to vibration detection and describe the required application. Next, we describe the processing pipeline and detail the detection statistics used to determine the presence of drilling vibrations. Multiple experiments at road and highway speeds show that the radar-based drilling detection system is able to raise an alarm based on drilling even when attempting to detect weak drilling vibrations in a vehicle going 112 kph. This work marks the first time that the millimeter-wave FMCW radar has been used for the alarming of intrusive drilling vibrations while in transport and shows promise to be included in multi-sensor security systems.

## 2. Background Information

### 2.1. Container Vibrational Model

Figure 1 shows the concept of the studied problem. Previous works have shown that it is possible to detect the vibration of a stationary container due to drilling using millimeter-wave radar. On a stationary container, Ref. [7] showed that drilling creates a micron-scale sinusoidal vibration in the container walls. 

If the drill rotates in the metal container at a frequency fdrill, then the drill will excite vibrations at the fundamental and harmonic frequencies of kfdrill/Nflutes, k∈ℤ+, as indicated in Figure 1b and visible in the blue line of Figure 2a. Nflutes is the number of sharp cutting flutes in the drill bit, typically two. The largest power is contained in the k=1 peak. The vibration influence of the drill on the container is denoted as vcd(t). The container used in experiments in this paper is a cylindrical steel container with dimensions labelled in Figure 1b.

As it moves along the road, the transport vehicle creates vibrations in both the container and radar sensor. These vibrations are problematic because they are anywhere from the millimeter to centimeter range in amplitude and easily drown out the micron-scale drilling vibrations. It is critical that we develop a statistical model for transport vibrations to gain a physical understanding of the detection problem and its limits. The transport vibrations are modeled as slow impulsive vibrations, denoted as $v_{ct}\left(t\right)$ as the influence of the transport on the container. The analogous vibration measured on the radar due to the transport is denoted as $v_{rt}\left(t\right)$. The physical meaning behind this choice is that the largest vibrations are from the transport vehicle encountering a discontinuity in the road construction and they experimentally show up as a sharp spike in the vibrational profile of the container, as measured by the sensor. 

From experiments, we have discovered that the impulsive transport vibration $v_{t}\left(t\right)$ tends to be of lower magnitude when the drilling $v_{d}\left(t\right)$ is present compared to when the drilling $v_{d}\left(t\right)$ is not present. This modulation is indicated in the mathematical model (1) as the indicator variable $\alpha 1_{d}$ (α∈[0,1]) and the arrow between Drill and Transport in Figure 1b. The physical meaning behind α is that, in order to drill through a thick container, the drill must impart a significant radial force into the container’s walls. This force is a restoring force, which somewhat stabilizes the container against impulsive vibrations.
(1)$$v_{r}\left(t\right)=v_{rt}\left(t\right)+\left(1-1_{d}\alpha \right)v_{ct}\left(t\right)+v_{cd}\left(t\right)$$

In (1), $v_{r}\left(t\right)$ is the total radial vibration as seen by the radar. The problem of detecting the drilling vibration is equivalent to isolating $v_{cd}\left(t\right)$ given $v_{r}\left(t\right)$. Immediately, we may choose the radar sensor’s position such that $v_{rt}\left(t\right)$ has a relatively low magnitude. As (1) is only concerned with radial vibrations, the choice to mount the radar on a plane with strength in the radial direction (as in Figure 1a) can drastically reduce $v_{rt}\left(t\right)$. Inserting the mathematical models for $v_{rt}$, $v_{ct}$, and $v_{cd}$ into (1), we can transform (1) into a statistical model of the total vibration $v_{r}\left(t_{s}\right)$ (2):(2)$$\hat{v_{r}}\left(t_{s}\right)=\underset{i}{\sum }\left(1-1_{d}\alpha \right)m_{i}p\left[a_{i}\left(t_{s}-\tau_{i}\right)\right]+\underset{k}{\sum }A_{k}\sin\left(\frac{2\pi kf_{drill}t_{s}}{N_{flutes}}\right)+n\left(t_{s}\right)$$

Using the statistical model in (2), we perform Monte Carlo simulations to fit all parameters to approximate experimental data with and without drilling, as seen in Figure 2. These results are shown in Table 1. 

In developing (2), we combine $v_{rt}\left(t\right)$ and $v_{ct}\left(t\right)$ into a single impulse generator (the first right-hand side term) because their contributions are inseparable. The variable $m_{i}$ is a normally distributed scalar corresponding to the magnitude of the i-th vibration impulse. p(t) is the fundamental vibration waveform from a transport-induced impulse at t=0; it is modeled as a ringing exponential, where the mean decay parameter $T_{p}$ and resonance frequency $f_{r}$ are determined from fitting experimental data, and the step function u(t) ensures that p(t)=0 for t<0. $\tau_{i}$ is the time of the i-th transport impulse where the differences in adjacent pulse times are well-modeled by a normal random variable. $A_{k}$ is the magnitude of the k-th drilling harmonic, which implicitly contains the transfer function from excitation point (Figure 1b) to the radar’s measuring point. 

From Table 1, we can conclude that the most problematic statistical model parameters are $m_{i}$ and $f_{r}$. If $m_{i}$’s values are too large (large standard deviation or mean), the vibration impulse waveforms will drown out all vibration. If $f_{r}$ is too similar to $f_{drill}/N_{flutes}$, then the vibration impulse spectrum will contain significant energy overlapping the main drilling harmonic. 

### 2.2. Millimeter-Wave Radar Operating Principle

In this section, we examine how to estimate the vibration on the container due to the drill $v_{cd}\left(t_{s}\right)$, as in (1), using a millimeter-wave radar sensor. To estimate $v_{cd}\left(t_{s}\right)$, we must first collect $v_{r}\left(t_{s}\right)$, the vibration seen by the radar, from the radar raw data. The process to recover $v_{r}\left(t_{s}\right)$ is diagrammed in Figure 3. For all experiments and analysis, we employ the commercially available AWR1642 millimeter-wave radar [8]. The AWR1642 operates from 77 to 81 GHz with an instantaneous bandwidth of 4 GHz. The in-depth settings for the AWR1642 are enumerated at the end of Section 4.

We consider a heterodyne FMCW millimeter-wave radar emitting a linearly ramped electromagnetic chirp at a slow-time $t_{s}$, as in (3) [9]:(3)$$x\left(t_{f},t_{s}\right)=\exp\left[j2\pi t_{f}\left(f_{L}+\frac{B}{T_{c}}t_{f}\right)\right]$$
where $t_{f}$ refers to fast time ($t_{f}=0$ at chirp start, $t_{f}=T_{c}$ at chirp end), $f_{L}$ is the lower operating frequency, B is the bandwidth of the chirp such that $f_{L}+B=f_{u}$, the upper operating frequency. $T_{c}$ is the total chirp time such that $B/T_{c}$ is the chirp slope in Hz/s. The electromagnetic chirp is transmitted and reflected, undergoing a phase shift related to the distance traveled (d) in the air. If the chirp is transmitted at a slow time of $t_{s}$, this phase shift contains the vibration information $v_{r}\left(t_{s}\right)$. The received chirp is then down-mixed with the transmitted chirp to produce an intermediate frequency (IF) waveform named $x_{IF}\left(t_{f},t_{s},rx\right)$:(4)$$x_{IF}\left(t_{f},t_{s},rx\right)=\exp\left[j2\pi t_{f}\left(\frac{B}{T_{c}}\right)\left(\frac{2d}{c}+\frac{2v_{r}\left(t_{s}\right)}{c}\right)-\frac{4\pi d}{\lambda_{c}}-\frac{4\pi v_{r}\left(t_{s}\right)}{\lambda_{c}}-\phi_{rx}\right]$$

If in (4) we assume that $d\gg v_{r}(t_{s}$), then $x_{IF}$ is a sinusoid oscillating at a beat frequency of $f_{b}=2Bd/cT_{c}$, which corresponds to a range of d. If d is constant, then the sinusoidal $x_{IF}$ has a phase (relative to a constant offset) of $4\pi v_{r}\left(t_{s}\right)/\lambda_{c}$. By collecting many received chirps over a slow time, we can track the phase of the $f_{b}$-component of $x_{IF}$ to reconstruct the vibration profile $v_{r}\left(t_{s}\right)$. In (4), we simplify the angle-of-arrival phase term as $\phi_{rx}$ for brevity, but $\phi_{rx}$ is a receiver-dependent phase term that corresponds to the angle-of-arrival of the reflection coming from d. Note that we assume that d is constant versus time—the container’s average position is stationary. This assumption is fair for large heavy containers, but a robust vibration estimation method would also track the mean distance to the container d over time.

To calculate the radar-seen vibration $v_{r}\left(t_{s}\right)$, the simplest method is to collect a number of chirps along slow time into a matrix frame. First, a range-FFT is performed (5); this transforms the fast-time axis ($t_{f}$) into its frequency counterpart, which is linearly related to the target range r:(5)$${\hat{X}}_{IF}\left(r,t_{s},rx\right)={ℱ}_{t_{f}}\left\{x_{IF}\left(t_{f},\;t_{s},rx\right)\right\}$$

Then, an angle-FFT is performed in (6) over the receiver dimension in order to spatially locate the target. The angle-FFT converts the receiver dimension (rx) to the angle dimension (θ). The AWR1642 radar only has an aperture along the azimuth plane, so the angle-FFT only resolves targets in azimuth, not elevation.
(6)$${\hat{\hat{X}}}_{IF}\left(r,t_{s},\;\theta \right)={ℱ}_{rx}\left\{{\hat{X}}_{IF}\left(r,t_{s},rx\right)\right\}$$

From ${\hat{\hat{X}}}_{IF}\left(r,\;t_{s},\theta \right)$, we must decide the range-and-angle (rangle) bins to track with the sensor. The tracking algorithm is implemented by first finding the rangle bin with a maximum energy at $t_{s}=0$ [10]. Then, the rangle bin is updated for each new $t_{s}$ after passing through a stability filter. The stability filter is a simple method that limits the maximum Cartesian distance that the chosen rangle bin can move between consecutive $t_{s}$ samples. The inclusion of the stability filter reduces the number of glitches that the millimeter-wave radar sensor sees during heavy vibrations. The chosen rangle bin is termed ($r_{m},\theta_{m})$.
(7)vr(ts)=λc4πtan−1(Im[X^^IF(rm,θm)] Re[X^^IF(rm,θm)])

The collection of $v_{r}\left(t_{s}\right)$ for many $t_{s}$ constructs the vibration profile. In practice, we drop the constant in front of (7) and unwrap the phase to allow for a larger measurement range, assuming that $t_{s}$ is fine-grained enough to capture all major vibration behavior. An example measurement of $v_{r}\left(t_{s}\right)$ and its spectrum was previously shown in Figure 2. 

## 3. Signal-Processing Pipeline

Given the vibration vector $v_{r}\left(t_{s}\right)$ measured from the millimeter-wave radar, the problem of detecting the presence of drilling is reduced to a simple hypothesis test:
$H_{0}:\;v_{r}\left(t_{s}\right)$ does not contain drilling $v_{cd}\left(t_{s}\right)$;$H_{a}$: $v_{r}\left(t_{s}\right)$ contains drilling$\;v_{cd}\left(t_{s}\right)$

Examining the statistical model built in (2), we decide that the best method of determining the presence of drilling is to look for constant peaks in the vibration spectrum versus time. By looking for constant peaks in the vibration spectrum, we are taking advantage of the fact that drilling will excite a small but constant frequency vibration while the vibrations from transport are larger but do not last for a long time or contain a single frequency. Detecting the presence of drilling by examining the PSD “baseline” in Figure 2b was another promising idea, but the addition of potentially non-stationary phase noise and the broadband impulsive nature of the driving-induced vibration leads this approach to result in a non-robust detection statistic.

This section will examine in depth the signal-processing pipeline required to precondition the vibration vector $v_{r}\left(t_{s}\right)$ for drilling detection. The entire pipeline from vibration to detection is diagrammed in Figure 4.

### 3.1. Time–Frequency Conversion

The first step is to convert the millimeter-wave radar output vibration vector $v_{r}\left(t_{s}\right)$ into a time–frequency representation ${\hat{V}}_{r}(t_{s},\;f_{s}$). The time–frequency representation means that at slow time $t_{s}$, the millimeter-wave sensor measured vibrational frequencies $f_{s}$ with magnitude ${\hat{V}}_{r}$. To perform time–frequency conversion, we compare three different algorithms on a dataset generated by measuring the drilling vibrations while moving: Short-Time Fourier Transform (STFT) [11], Continuous Wavelet Transform (CWT) [12,13,14,15], and Adaptive Super-Resolution Wavelet Transform (ASLT) [16] below in Figure 5. 

In Figure 5, we examine the time–frequency conversion of a measured vibration vector from an experiment that begins with drilling-while-driving for a duration of 6.5 s. The drill used is rated for 2000 rpm, which converts to 33.3 Hz. The drill-bit is a two-flute 6.35 mm-diameter titanium drill-bit, which leads us to expect a fundamental vibration frequency of 16.7 Hz. In Figure 5b–d, this harmonic is clearly visible as the horizontal line extending from 0 to approximately 6 s. The STFT, CWT, and ASLT algorithms all correctly identify the major harmonic from which we perform the detection. The ASLT-generated spectrogram in Figure 5d shows a major reduction in spectral noise compared to the STFT, and especially the CWT. However, detection performance for the ASLT-generated spectrogram is poor due to the low magnitude of the 16.7 Hz harmonic relative to the non-drilling noise (−15 dB relative maxima). Additionally, ASLT is computationally expensive compared to both STFT and CWT. The STFT suffers from the same magnitude issue, where even though the STFT-generated spectrogram (Figure 5b) has a large, consistent time–frequency drilling harmonic, the peak magnitude of the 16.7 Hz horizontal bar relative to the non-drilling noise (−14.6 dB max) affected detection performance. The CWT-generated spectrogram (Figure 5c), though it appears the noisiest, results in the highest magnitude ratio of all three algorithms tested (−6.7 dB), which implies the best detection performance. 

Therefore, we chose to convert $v_{r}\left(t_{s}\right)$ into its time–frequency conversion using the Continuous Wavelet Transform (CWT) as in (8):(8)$${\hat{V}}_{r}\left(t_{s},\;f_{s}\right)=C\left\{v_{r}\left(t_{s}\right)\right\}$$
where C{…} refers to the CWT operator. To implement the CWT, we utilized MATLAB’s wavelet toolbox [15]. We optimized the CWT parameters for detection by using a bump wavelet and 48 voices per octave. 

### 3.2. De-Cluttering

After the vibration’s time–frequency representation is obtained using (8), the spectrogram must still be de-cluttered because it contains the clutter from impulsive vibrations and sensor artifacts. The first step is to first divide each chirp by its total energy (9):(9)$${\hat{V}}_{r,n}\left(t_{s},f_{s}\right)=\frac{{\hat{V}}_{r}\left(t_{s},f_{s}\right)}{\int_{f_{lo}}^{f_{hi}}{\left|{\hat{V}}_{r}\left(t_{s},f_{s}\right)\right|}^{2}df_{s}}$$
where, now, the subscript n indicates energy normalization. In this work, $f_{lo}=1$ Hz and $f_{hi}=50$ Hz. This major step ensures that the high-energy impulsive behavior as modeled in (2) is penalized. The rationale behind this is that, because drilling both excites a weak oscillation and dampens the high-energy impulsive driving-induced strikes, a chirp that experiences a high-energy impulse is likely to be a non-drilling chirp. By penalizing the total energy of a chirp, these high-energy impulses cease to dominate the image (as in Figure 5c and Figure 6a). 

Next, we perform a 2-D median absolute deviation (MAD) detection to enhance the peaks in the spectrogram [17,18]. First, a local 2-D image median is calculated and then used to calculate the local 2-D MAD. In (10), we describe this process using the median filter operator ${ℳ}_{ab}\left\{\ldots \right\}$. ${ℳ}_{ab}\left\{A\right\}$ will take a matrix A and replace each A(i,j) pixel by the median of a local area centered around the (i,j) pixel in the shape of a rectangle of size a×b.
(10)$${\hat{V}}_{r,n,m}\left(t_{s},f_{s}\right)=1.4826\cdot {ℳ}_{ab}\left\{{\hat{V}}_{r,n}-{ℳ}_{ab}\left\{{\hat{V}}_{r,n}\right\}\right\}$$
where, now, the subscript m indicates that the vibration spectrogram has been median-filtered. In (10), MAD is used in place of a typical Z-score because ${\hat{V}}_{r,n}$ still contains peaks large enough to self-bias the resulting value of ${\hat{V}}_{r,n,m}$. Using MAD instead allows the horizontal peaks to be properly adjusted. We find a value of a = 32, b = 32 to obtain acceptable results. This results in the 2-D MAD being calculated over a local region of approximately 3.3 Hz and 8.6 ms of slow time. The operator ${ℳ}_{ab}\left\{\ldots \right\}$ is implemented using MATLAB’s medfilt2 function.

The effect of 2-D MAD filtering on the resulting image is more subtle (as in Figure 6b) than energy normalization but similarly important. By calculating the relative importance of each pixel, ${\hat{V}}_{r,n,m}$ has emphasized consistent peaks in the image and de-emphasized noise, which increases the detection capability of the processing pipeline. In terms of computation time, the implementation of (9) takes the longest of all steps. For this reason, b can be tuned downwards to decrease computation time.

### 3.3. Search for Horizontal Harmonics

Now that ${\hat{V}}_{r,n,m}$ has had its energy normalized and peaks emphasized, we further select horizontal frequency stripes by performing two steps. First, a single-pass horizontal line filter is performed. Second, we apply principal component analysis (PCA) to the image to better find the horizontal stripes indicative of the presence of drilling.

The horizontal line filter is implemented by convolving ${\hat{V}}_{r,n,m}$ with a horizontal stripe kernel as in (11) [19,20]:(11)$${\hat{V}}_{r,n,m,f}={\hat{V}}_{r,n,m}*H_{c}$$
where, now, the subscript f means that the spectrogram has been horizontal-line-filtered, the operator ∗ refers to 2-D convolution, and $H_{c}$ is a horizontal line filter of width c, as in (12):(12)$$H_{c}=\left[\begin{array}{ccc} -1 & -1 & -1\\ \begin{array}{c} 2/c\\ \vdots \\ 2/c \end{array} & \begin{array}{c} 2/c\\ \vdots \\ 2/c \end{array} & \begin{array}{c} 2/c\\ \vdots \\ 2/c \end{array}\\ -1 & -1 & -1 \end{array}\right]$$

In (12), c refers to the number of positive elements in the row dimension. c is tuned to be the approximate pixel width of the drilling harmonic. From measurement, we chose c=16. The choice of c too low can split and reduce a drilling harmonic, and the choice of c too high can spread out the drilling harmonic over $f_{s}$, leading to biased statistics once detection is performed.

The inclusion of the horizontal line filter significantly decreases the amount of non-horizontal clutter present in ${\hat{V}}_{r,n,m}$. Further, (11) decimates non-horizontal peaks emphasized by the 2-D MAD processing step (10). However, short-lived horizontal peaks, which are not indicative of drilling but rather long-ringing transport vibrations, are not entirely removed by (11). For this reason, we apply our final pre-processing step of PCA. 

To perform PCA [21,22,23] on the spectrogram, ${\hat{V}}_{r,n,m,f}$ is first decomposed into its singular value decomposition (SVD) representation in (13).
(13)$$U\Sigma V^{T}={\hat{V}}_{r,n,m,f}\left(t_{s},\;f_{s}\right)$$

In (13), U is comprised by the left singular vectors, V is comprised by the right singular vectors, and Σ is a diagonal matrix with singular values on the diagonal. The principal of our PCA application is that the left vectors make up the eigenvectors of its autocorrelation matrix ${\hat{V}}_{r,n,m,f}{\hat{V}}_{r,n,m,f}^{T}$. A strong horizontal stripe in ${\hat{V}}_{r,n,m,f}$ will result in a strong eigenvalue of ${\hat{V}}_{r,n,m,f}{\hat{V}}_{r,n,m,f}^{T}$, which is equal to the square of the corresponding singular value in Σ. Therefore, to search for horizontal stripes in ${\hat{V}}_{r,n,m,f}$, we look for the largest L singular values of its SVD and then reconstruct the spectrogram, keeping only the largest L singular values and vectors. An example of the singular values from a decomposition of the example data from Figure 6 after horizontal line convolution is shown in Figure 7.

In Figure 7b–d, SI N refers to the image generated using only the N-th singular value. The first singular image (SI 1) contains the majority of the horizontal striping, while SI 2 and SI 3 contain smaller-magnitude variations around the horizontal stripes. Therefore, the most aggressive reconstruction uses only the first singular image. To ensure robustness, however, we choose the number of images L simply by thresholding the magnitude of the singular values. From experiments, we have found that excluding all singular values with a magnitude of less than half of the maximum singular value provides acceptable results. In Figure 7a, this means that we take L=1. Once L is determined via this process, the spectrogram is reconstructed using (14):(14)$${\hat{V}}_{r,n,m,f,p}\left(t_{s},\;f_{s}\right)≜I\left(t_{s},f_{s}\right)=U_{L}\Sigma_{L}V_{L}^{T}$$
where $U_{L},\Sigma_{L},$ and $V_{L}$ are the SVD-components of ${\hat{V}}_{r,n,m,f}$ with only the first L singular vectors and values. Additionally, negative values of the resulting spectrogram are set to zero. The subscript p of ${\hat{V}}_{r,n,m,f,p}$ implies that the image has undergone PCA. As PCA is the final pre-processing step, ${\hat{V}}_{r,n,m,f,p}$ is re-named as I for brevity. Now, $I\left(t_{s},f_{s}\right)$ is ready for a statistical analysis.

### 3.4. Detection and Probability-Versus-Time 

To quantify how likely it is that a given slow-time bin ts contains drilling, we must first create a detection statistic T(ts): ℝNts×Nfs→ℝNts. As negative spectrogram values have been zeroed out in the previous PCA step, the histogram of a linearly collapsed I(ts,fs) resembles a one-sided heavy-tailed distribution. To quantify the significance of a given pixel (ts,fs), we divide it by the overall standard deviation of the spectrogram. Then, we take the column-wise maximum to collapse the Nts×Nfs matrix into a vector representing the maximum detection statistic over all frequencies for a given slow-time bin. The detection statistic T(ts) is illustrated in (15):(15)T(ts)=maxfsI(ts,fs)std{I}
where std{I} refers to the standard deviation over all pixels in the spectrogram. Under the null hypothesis, we approximate that the statistic T(ts) follows a half-normal distribution. An example T(ts) is shown in Figure 8a. This assumption is fair because the myriad of pre-processing steps strongly filtered out the impulsive vibrations, causing the remaining pixels in a non-drilling scenario to be mostly noise. Then, we can quantify the probability that T(ts) is anomalous by examining the standard normal cumulative density function (CDF) in (16):(16)P^drill(ts)=Φ[T(ts)]−Φ[−T(ts)]
where Φ(z) is the standard-normal CDF evaluated at point z. An example of P^drill(ts) is shown in the red curve of Figure 8b.

One last step is required before claiming that drilling is detected. As the maximum operator in (15) is prone to quantitative errors, we must perform a stabilizing of the probability. Stabilizing is conducted by taking the product of the previous probability, current probability, and future probability of drilling as in (17):(17)$$P_{drill}\left[n\right]=\overset{n+1}{\underset{i=n-1}{\prod }}{\hat{P}}_{drill}\left[i\right]$$
where the continuous subscript has been dropped for the discrete subscript [n] for the n-th slow-time sample. The removal of the hat from $\hat{P}$ symbolizes that $P_{drill}$ is now time-stable. An example of $P_{drill}$ can be seen calculated on experimental data in the black curve of Figure 8b.

Finally, at this point, a decision can be made about the presence of drilling or not. Based on the duration of above-threshold probability, an alarm will sound. The method used is whether 90% of any continuous one-second interval is alarming. This is calculated using an integral over consecutive one-second periods and by checking whether the result is greater (alarm) or less (no alarm) than 0.9. Once the alarm is sounded, the alarm is latched until sensing a one-second period without over 90% probability of drilling. Naturally, the thresholds may be adjusted up or down in order to decrease the false alarm rate.

## 4. Experimental Setup

To validate the millimeter-wave radar sensor’s capabilities, we drove with a cylindrical steel transport container placed in the back of a car along a state highway in California, indicated in Figure 9. The AWR1642 radar was pointed directly at the container from a distance of 0.5 m. A 2000 rpm drill with a 6.35 mm diameter drill-bit was positioned on the opposite side of the radar so that the AWR1642 would not detect the drill’s vibrations directly. Naturally, if the drill was visible to the radar, the same processing pipeline easily works with increased power. For rigor, we collected datasets both with and without drilling present. Drilling-absent datasets were artificially inserted in the middle of larger drilling-present datasets to test the responsiveness and limits of the signal-processing pipeline. The car drove at a steady 112 kph and all datasets were saved to a laptop PC for later processing in MATLAB. 

In Figure 9, the AWR1642 is mounted securely on the side of a tripod laser Doppler vibrometer (LDV). The LDV was used to verify that the vibration spectra seen by the AWR1642 sensor were correct. The AWR1642 was programmed with the settings described in Table 2.

The settings result in a maximum range of 24.6 m and a slow-time sampling rate of 3.7 kHz for a maximum un-aliased vibration frequency of 1.85 kHz. Although we discarded vibrations above 50 Hz and typically operated less than a meter from the container, we found this combination of settings to result in stable sensor behavior. A custom sensor solution may benefit from slower slow-time sampling rates.

## 5. Results

In Figure 10, we present three fully processed datasets following the format of drilling–no drilling–drilling. The difference between drilling and no drilling data is drawn using a vertical dashed black line. An alarm is triggered if the drilling probabilities hold above 90% total for a continuous second, as described in the previous section. Sections when the alarm was triggered are highlighted in red, while sections when the alarm was lowered are highlighted in green. Text indicating the alarm status is placed in each section.

The first dataset, displayed in Figure 10a, correctly identifies and alarms at the presence of drilling during all three experimental sections. Out of all three datasets, however, this dataset experienced a unique low probability of detection after 3.2 s. The low probability causes the alarm to be dropped even while drilling is still known to be occurring. Given that the alarm was raised earlier in the dataset and from examining the un-processed spectrograms, it appears that the $t_{s}\in [3.2,\;5.3$] s time period in Figure 10a experienced an abnormally high amount of impulsive vibration, which is consistent with driving over a rough stretch of highway. As (9) normalizes the spectrogram with respect to column-wise energy, the energy from drilling in this time period was drowned out. From this phenomenon, we can conclude that the proposed signal-processing pipeline will not cause false alarms from rough stretches of highways but will be insensitive to drilling during those times. As long as the drilling occurs with at least one second of a relatively smooth ride, the proposed pipeline can raise the alarm. In the statistical model (2), this phenomenon is related to increasing the average values of the impulsive peak magnitude m.

The second and third datasets, Figure 10b,c both show excellent detection behavior and quality, correctly identifying and alarming all three experimental sections. A phenomenon that we noticed in both Figure 10b,c is the slow ramping behavior of probability from drilling to non-drilling sections. This behavior occurs for several reasons. First, the CWT will spread the 16.7 Hz drilling harmonic into the non-drilling section due to a limited time–frequency resolution. The times near where the drilling starts and stops will be of lower magnitude compared to the center of the horizontal harmonic. The horizontal line convolutional filter and PCA will further smear the weakened drilling harmonic into the non-drilling section. Therefore, upon detection analysis, the borders near drilling and non-drilling regions contain a weaker version of the 16.7 Hz harmonic, which will result in a low probability of detection near edges, as seen in Figure 10b,c. Finally, (17) ensures that $P_{drill}$ is correlated with its past and future states, which means that it cannot change instantaneously. 

The last phenomenon seen in the experimental data is the sizeable peak in the non-drilling zone of Figure 10c. Examining the raw spectrogram, the region around 11 s in the non-drilling zone contains small but visible horizontal spectral peaks at 14 Hz and 20 Hz that live for approximately 1.5 s. These peaks are close enough to the 16.7 Hz peak that they are not entirely removed by PCA, and we see a small peak in the probability of drilling at around 10.5 s in Figure 10c. In the time instable probability (16), the probability of the dataset behind Figure 10c peaks at 0.64, but the stability filter (17) reduces this to a peak of 0.27. From this phenomenon, we can conclude that the main sensitivity of the proposed signal-processing pipeline is the presence of random harmonic vibrations that look exactly like drilling, i.e., $f_{r}$ in (2) is similar to $f_{drill}/N_{flutes}$. The proposed algorithm is mostly robust against this behavior, but certain conditions (such as high magnitude and proximity to main drilling harmonic) make this difficult to remove. Luckily, a deployment with large harmonic vibrations near the drilling frequency may deal with this problem by simply re-structuring the mechanical layout as the vibrations are a function of the container and its mounting. 

## 6. Conclusions

In this paper, we proposed a millimeter-wave radar-based sensor and signal-processing pipeline intended to alarm if drilling is sensed on a transport container. Due to its short wavelength, a millimeter-wave radar can sense the micron-scale vibrations excited in transport containers due to intrusive drilling. However, the container experiences high-magnitude vibration from transport, which tends to drown out the low-magnitude vibration from drilling. Beginning with the raw data, we described the methods used to track the container and extract the vibration profile. From the vibration profile, we proposed a novel pipeline to examine the spectrogram and heavily pre-process this to get rid of the extremely intrusive, impulsive noise to create a robust detection statistic. From this detection statistic, we described an alarm decision that performs extremely well in a test environment. In future research, it would be interesting to quantify the sensor performance over entire application lifetimes and for a variety of container geometries. Overall, we have found millimeter-wave radar-based sensors to be excellent candidates for a relatively inexpensive and scalable detection of intrusive drilling during transport.

## Figures and Tables

**Figure 1 sensors-22-02718-f001:**
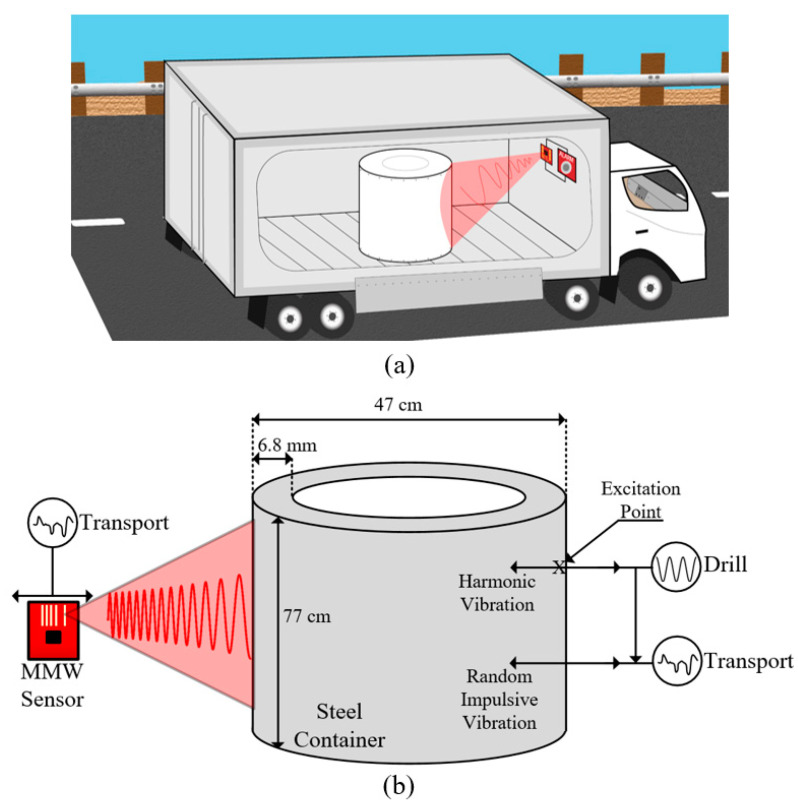
(**a**) Concept of drilling detection during container transport, (**b**) vibrational model with harmonic and random impulsive vibrations.

**Figure 2 sensors-22-02718-f002:**
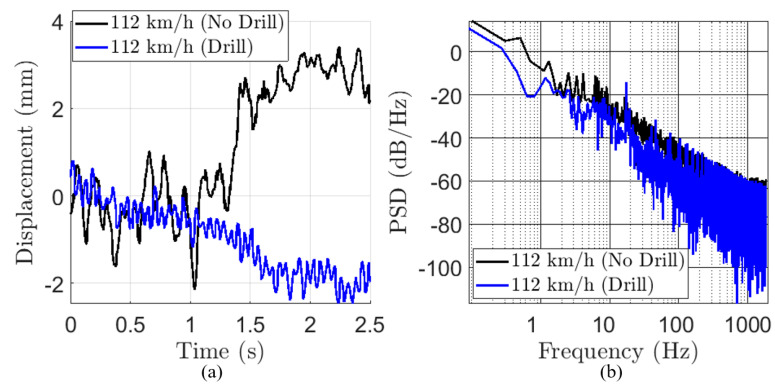
(**a**) Measured time-domain vibration waveform $v_{r}\left(t\right)$ with (blue) and without (black) drilling measured at 112 km/h on a highway. (**b**) Power spectral density of (**a**). Measurement duration is approximately 6 s, zoomed in.

**Figure 3 sensors-22-02718-f003:**
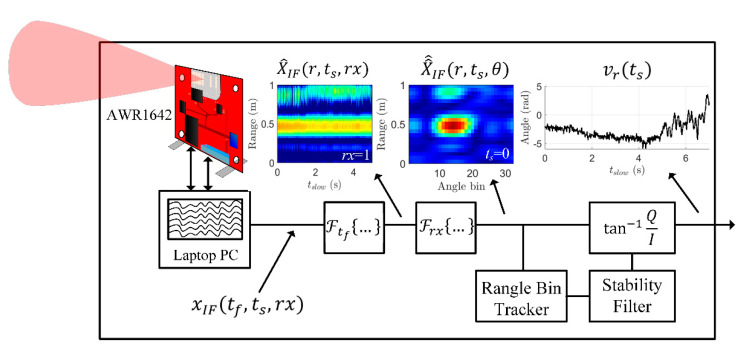
Block diagram of processing steps required to extract a stable vibration measurement vector from the radar. Each node is labeled with the corresponding variable and equation. ${\hat{\hat{X}}}_{IF}\left(r,t_{s},\theta \right)$ is padded to 32 angle bins from 4 for display purposes.

**Figure 4 sensors-22-02718-f004:**
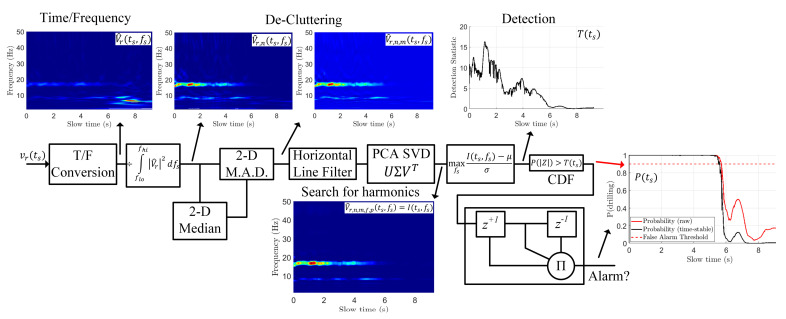
Pre-processing pipeline block diagram in order to turn $v_{r}\left(t_{s}\right)$ into a detection. Major steps have the resulting spectrogram shown after the process, with labels for various stages of pre-processing. On many images, the variable label used in this paper’s equations are included on the image itself.

**Figure 5 sensors-22-02718-f005:**
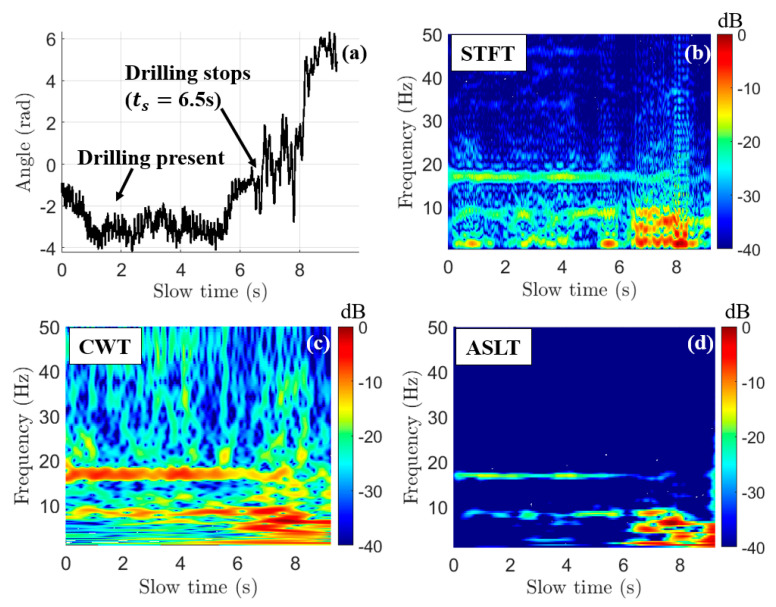
Comparison of time–frequency conversion algorithm performance when analyzing drilling-while-driving data. All spectrograms are magnitude-only in dB scale. (**a**) An example-measured vibration vector $v_{r}\left(t\right)$ measured in radians with drilling present up to 6.5 s. (**b**) Short-Time Fourier Transform (STFT) spectrogram (window length of 2048 chirps). (**c**) Continuous 1-D Wavelet Transform (CWT) spectrogram (bump wavelet, 48 voices/octave). (**d**) Adaptive Super-Resolution Wavelet Transform (ASLT) generated using reference code in [16] (Morlet, $N_{cyc}$ = 6).

**Figure 6 sensors-22-02718-f006:**
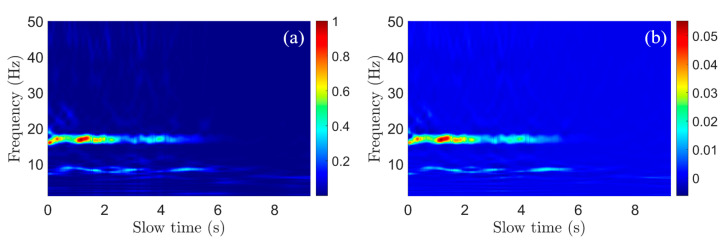
(**a**) Spectrogram after energy normalization (8); high-magnitude impulsive peaks in the spectrogram (Figure 5c) are decimated. (**b**) Spectrogram after 2-D median absolute deviation (MAD) (9). Both spectrograms are displayed in linear scale.

**Figure 7 sensors-22-02718-f007:**
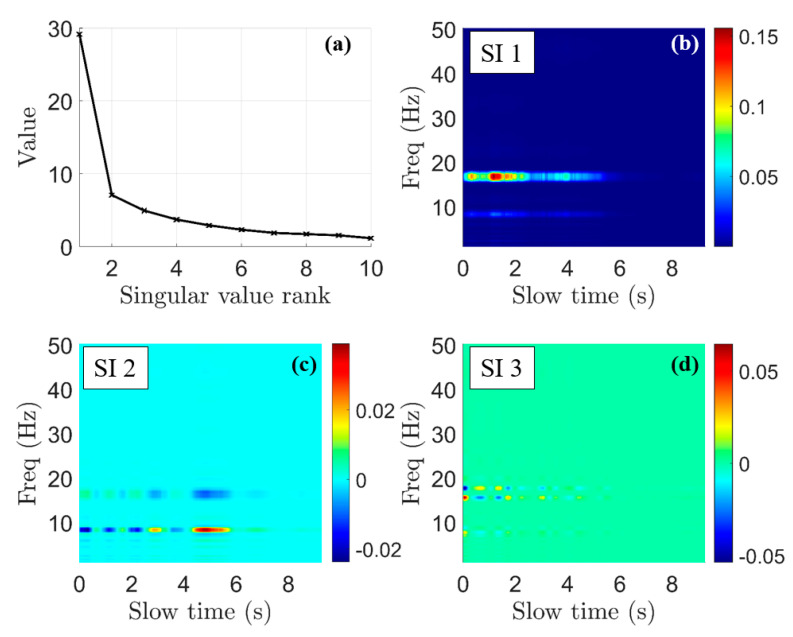
Singular values and images for explanation of principal component analysis (PCA) on experimental data. (**a**) Singular values ranked by magnitude. (**b**) Singular image reconstructed using only the first singular value (SI 1). (**c**) Singular image using the second singular value (SI 2). (**d**) Singular image using the third singular value (SI 3).

**Figure 8 sensors-22-02718-f008:**
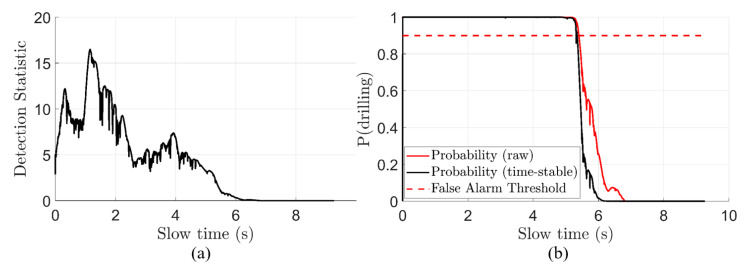
(**a**) Detection statistic $T\left(t_{s}\right)$ versus slow time, calculated using (15). (**b**) Probability of drilling, stable and non-stable versions, calculated using (16), red curve, and (17), black curve.

**Figure 9 sensors-22-02718-f009:**
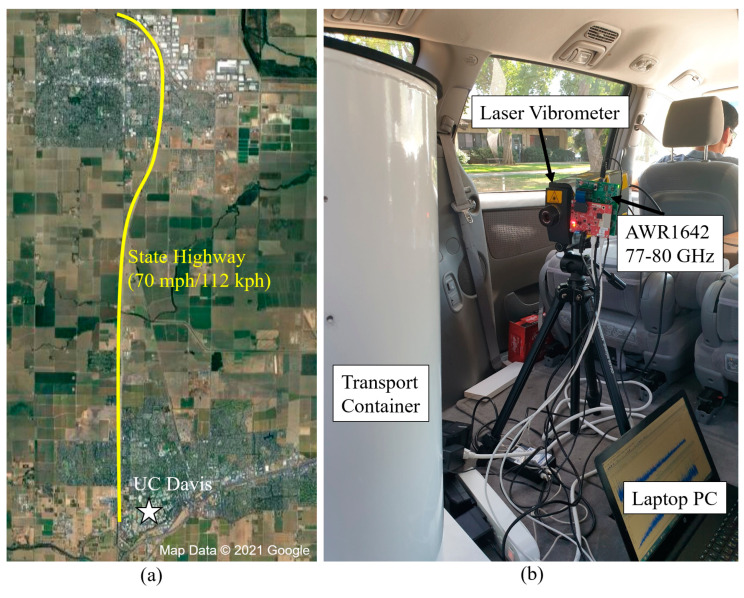
(**a**) Satellite view of roads traveled for container drilling-while-driving experiment. (**b**) Setup in the back of a van, radar mounted on stable tripod and connected to a laptop PC. A Laser Doppler Vibrometer (LDV) is included to verify vibration spectra.

**Figure 10 sensors-22-02718-f010:**
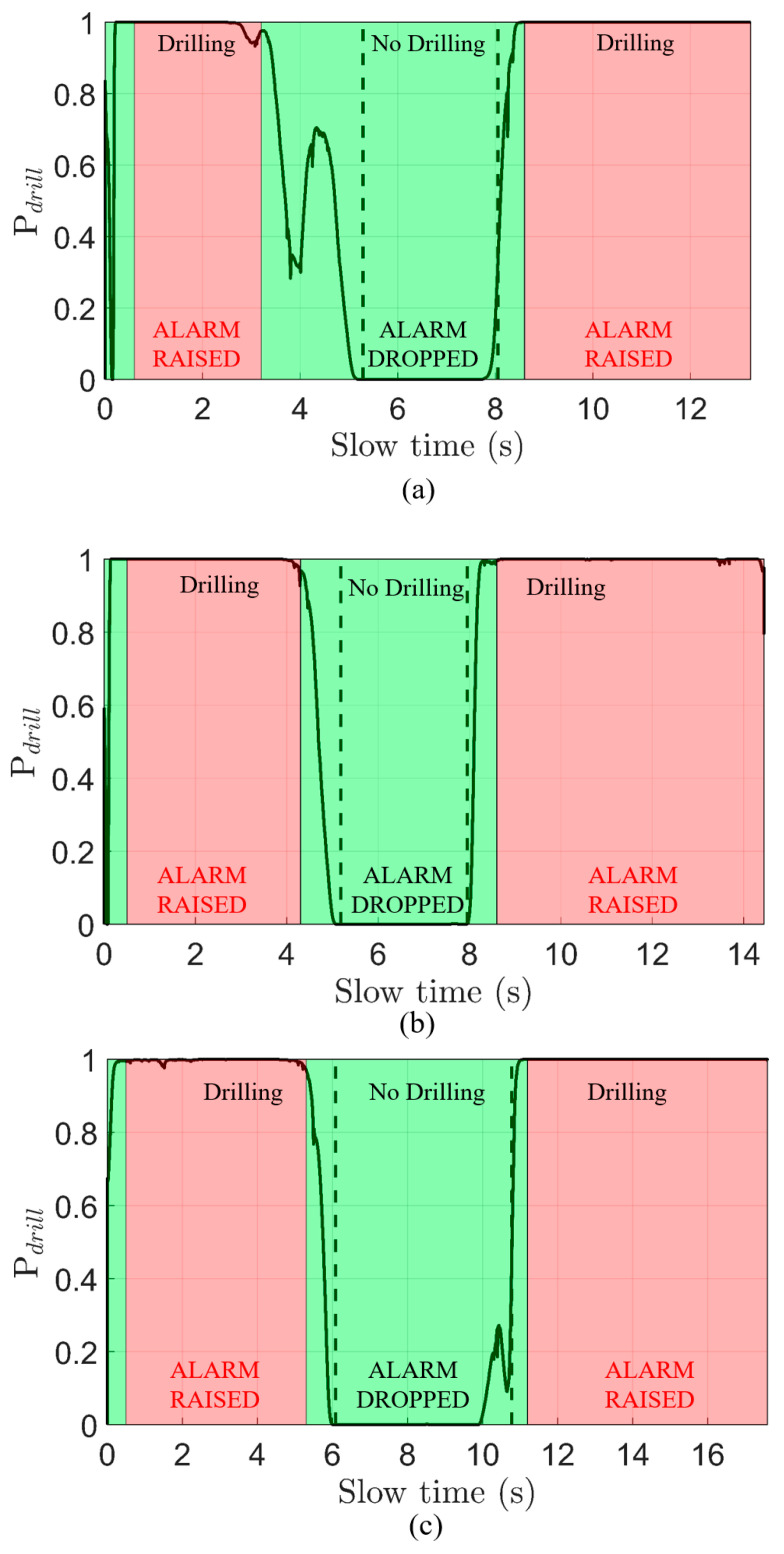
Processed and alarmed drilling datasets collected using the AWR1642 mmW radar sensor while driving at 112 kph on a highway. Each dataset consists of two periods of drilling separated by a period of no drilling. In each dataset, the black line represents the calculated probability of drilling from (17). A red highlight over a time section implies that an alarm has been raised and not lowered. A green highlight over a time section implies that the alarm has been lowered. (**a**) Dataset 1—total time of 13.2 s. (**b**) Dataset 2—total time of 14.4 s. (**c**) Dataset 3—total time of 17.6 s.

**Table 1 sensors-22-02718-t001:** Random variables and parameters in the statistical model of (2) with maximum likelihood estimators (MLEs) indicated, calculated from experimental conditions.

Parameter	Significance	112 km/h Experimental Estimate	Method of Determination
$m_{i}$	Magnitude of radar vibration due to transport (mm)	$m_{i}\sim N\left(\mu ,\sigma^{2}\right)$ $\hat{\mu }=-0.3,\;\hat{\sigma }=0.9$	Monitor impulsive peaks from experimental data
p(t)	Base vibration impulse waveform	$p\left(t\right)=\exp\left(-\frac{t}{T_{p}}\right)\cos\left(2\pi f_{r}t\right)u\left(t\right)$ $T_{p}\approx$$\;220\;\mathrm{ms},\;f_{r}\approx 5$ Hz	Fit waveform shape and spectrum to experimental data
$\tau_{i}$	Time of the *i* vibration impulse (s)	$\left(\tau_{i}-\tau_{i-1}\right)\sim N\left(\mu ,\sigma^{2}\right)$ $\hat{\mu }=0.24,\;\hat{\sigma }=0.12$	Monitor impulsive peaks from experimental data
α	Magnitude dampening from presence of drilling (0–1)	$\hat{\alpha }=$ 0.377	Monitor average PSD with and without drilling
$A_{k}$	Magnitude coefficient of the *k*th drilling harmonic	$\frac{A_{0}}{f_{drill}/N_{flutes}\left(k-1\right)\;},\;A_{0}=0.112$	Fit to experimental data $\left(f_{drill}=33.3\;\mathrm{Hz}\right)$
n	mmW-radar vibration estimation noise (mm)	$n\sim N\left(0,\;\sigma^{2}\right),\;\hat{\sigma }=3\cdot 10^{-3}$	Estimate from radar phase noise based on stationary target

**Table 2 sensors-22-02718-t002:** AWR1642 settings for vibration detection.

Parameter	Value	Parameter	Value
Chirp time (*T_c_*)	170 μs	Center frequency (*f_c_*)	78.6 GHz
Idle time $\left(T_{idle}\right)$	100 μs	Number of fast-time samples *N_t_f__*	256
Bandwidth (*B*)	3.23 GHz	Number of receivers	4
Chirp slope (*S*)	19 MHz/μs	Number of transmitters	1
ADC sampling rate	6.25 MS/s		

## Data Availability

This study did not report any open-source data.

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
