# Peer review of "Detecting the Presence of Intrusive Drilling in Secure Transport Containers Using Non-Contact Millimeter-Wave Radar"

_sensors, 2022, doi:10.3390/s22072718_

Round 1
Reviewer 1 Report
The article demonstrates that a 77-81 GHz Frequency-Modulated Continuous Wave (FMCW) millimeter-wave radar vibration sensor can be used to detect micron-scale intrusive drilling while vehicle movement shakes the container.
- The explanation for the motivation to use the system in the field of millimeter waves should be expanded in addition to the proposed explanation of commercial availability.
- In addition to statistical analysis, it is necessary to calculate the linear velocity of the tank and to extract the Doppler distraction from the calculation.
- It is advisable to explain why the authors chose the FMCW modeling kit over the CW radar system.
- The case presented is particular. It is advisable to expand the measurements for containers with different geometry and different contents.
Reviewer 2 Report
In general this paper is well written, though it could do with a really good English speaker to go through it and correct the occasional instances where the phrasing is curious. However, my main problem with the paper (and the research) is that the authors are examining a "toy"problem. The whole paper, and all of the finely tuned signal processing (which I cannot fault) is centered around a specific case of drilling into a particular cylindrical container with a specific height, diameter and wall thickness. In my view it is unlikely that this will ever be a common situation. Surely, if the authors wanted to provide more useful research, they would have focused their work onto identifying drilling into one of the ubiquitous 20 or 40 ft containers. If they were able to show that their algorithm is capable of detecting drilling into such a complex enclosure, then I would be impressed.
I suggest this paper needs to be extended to include such containers.
Reviewer 3 Report
The paper presents the research on an innovative approach of using non-contact millimetre-wave radar to monitor the safety of containers during the transport process. This process allowing to filter intrusive drilling micro-vibrations from the transport micro-vibrations is certainly a novel and original proposal of the research team. The presented paper certainly shall be considered an important attempt and it certainly addresses a certain gap in this field.
The research part is well-preceded by the Introduction based on relatively short and not-exhaustive, still sufficient and well-structured state-of-art report. Widening and updating of this part might make this interesting paper more beneficial to the reader.
The research supported with well-derived analytical part, itself is well-designed and executed, the methodology is carefully and clearly explained, the results are well described and concluded.
Authors do not draw a line towards new research directions or other uses of this inexpensive and very effective approach, which would be welcomed.
The paper is well-written in English, draws and tables are clear to the reader. The paper is edited on a high level.
Round 2
Reviewer 1 Report
The authors did not expand the data presented for containers with different geometry or different contents. The article deals with a particular case and is not included in a general problem.
Reviewer 2 Report
It is still a 'toy" problem, and a single comment in the Conclusions stating that tests with other containers would be useful does not cut it.
